# MicroRNAs as Indicators of Malignancy in Pancreatic Ductal Adenocarcinoma (PDAC) and Cystic Pancreatic Lesions

**DOI:** 10.3390/cells11152374

**Published:** 2022-08-02

**Authors:** Christian Prinz, Leonard Fehring, Robin Frese

**Affiliations:** Medizinische Klinik 2, Helios Universitätsklinikum Wuppertal, Lehrstuhl für Innere Medizin 1 der, University of Witten gGmbH, 42283 Wuppertal, Germany; leonard.fehring@helios-gesundheit.de (L.F.); robin.frese@helios-gesundheit.de (R.F.)

**Keywords:** microRNA, pancreatic cyst, pancreatic ductal adenocarcinoma, hyperinsulinemia, dysregulation, anti-apoptotic pathways

## Abstract

The dysregulation of microRNAs has recently been associated with cancer development and progression in pancreatic ductal adenocarcinoma (PDAC) and cystic pancreatic lesions. In solid pancreatic tumor tissue, the dysregulation of miR-146, miR-196a/b, miR-198, miR-217, miR-409, and miR-490, as well as miR-1290 has been investigated in tumor biopsies of patients with PDAC and was reported to predict cancer presence. However, the value of the predictive biomarkers may further be increased during clinical conditions suggesting cancer development such as hyperinsulinemia or onset of diabetes. In this specific context, the dysregulation of miR-486 and miR-196 in tumors has been observed in the tumor tissue of PDAC patients with newly diagnosed diabetes mellitus. Moreover, miR-1256 is dysregulated in pancreatic cancer, possibly due to the interaction with long non-coding RNA molecules that seem to affect cell-cycle control and diabetes manifestation in PDAC patients, and, thus, these three markers may be of special or “sentinel value”. In blood samples, Next-generation sequencing (NGS) has also identified a set of microRNAs (miR-20a, miR-31-5p, miR-24, miR-25, miR-99a, miR-185, and miR-191) that seem to differentiate patients with pancreatic cancer remarkably from healthy controls, but limited data exist in this context regarding the prediction of cancer presences and outcomes. In contrast to solid pancreatic tumors, in cystic pancreatic cancer lesions, as well as premalignant lesions (such as intraductal papillary neoplasia (IPMN) or mucinous-cystic adenomatous cysts (MCAC)), the dysregulation of a completely different expression panel of miR-31-5p, miR-483-5p, miR-99a-5p, and miR-375 has been found to be of high clinical value in differentiating benign from malignant lesions. Interestingly, signal transduction pathways associated with miR-dysregulation seem to be entirely different in patients with pancreatic cysts when compared to PDAC. Overall, the determination of these different dysregulation “panels” in solid tumors, pancreatic cysts, obtained via fine-needle aspirate biopsies and/or in blood samples at the onset or during the treatment of pancreatic diseases, seems to be a reasonable candidate approach for predicting cancer presence, cancer development, and even therapy responses.

## 1. Introduction

### MicroRNA Dysregulation Shows Different Patterns in Pancreatic Ductal Adenocarcinoma (PDAC) and Pancreatic Cysts (PC)

Pancreatic cancer is increasingly frequent in Western countries and among the top seven leading causes of cancer-related deaths worldwide [1]. Pancreatic cancer remains one of the most lethal malignant neoplasms, causing 432,242 new deaths in 2018 worldwide, with 355,317 new cases estimated to occur by 2040. Local infiltrative growth, peritoneal carcinosis, and liver metastases determine operability, and local resectability criteria are based in particular on tumor growth in the surrounding vessels [2]. Painless jaundice is a characteristic sign of pancreatic cancer, but it is frequently associated with an advanced tumor stage. In most cases of pancreatic cancer, characteristic early symptoms are missing; according to the American Cancer Society, only 9% of all pancreatic cancers are diagnosed at a localized tumor stage, while more than 50% of patients present with distant metastasis at diagnosis [3]. Therefore, treatment is often restricted to palliative regimes. Regardless of the underlying tumor stage, overall 5-year survival is very poor, with only about 4–6% of patients surviving after 5 years [3]. In general, malignant pancreatic tumors are categorized as follows: surgically resectable, locally advanced (LA), and metastatic pancreatic cancer. However, due to non-detectable distant metastasis and a high rate of local recurrences, the 5-year survival rate after R0 resection is about 20% [3]. Chemotherapy is increasingly used to treat locally advanced tumors, based on the high response rates of patients with metastatic pancreatic cancer after FOLFIRINOX chemotherapy [4]. Recent studies demonstrate that induction chemotherapy with FOLFIRINOX in combination with percutaneous radiotherapy is associated with a significant response [5].

In the clinical practice of the experienced interventional gastroenterologist, fine-needle aspiration of pancreatic tumor tissue or pancreatic cyst fluids is routinely performed to investigate the dignity of pancreatic processes and fluid-containing cysts. Due to the low output of cellular materials, the sensitivity and the determination of cancer presence are often very poor, with reported sensitivities of less than 80% in pancreatic ductal adenocarcinoma (PDAC) (PC) [6,7] and less than 50% in biliary tract cancers [8,9,10,11]. Classical risk profiles for the development of these cancer types include genetic disposition, alcohol consumption, smoking, and even repeated endoscopic retrograde cholangio-pancreatography (ERCP) examination with retrograde entry of bacteria into the biliary system. Furthermore, pancreatic cysts containing mainly fluids become neoplastic, with aggressive progression and systemic metastasis. This affects overall survival significantly [12,13,14]. However, from a clinical point of view, the approach to patients with pancreatic adenocarcinoma is, in most cases, delayed, so that the majority of patients are diagnosed when the tumor has frequently metastasized, and, despite modern treatment, surgery and chemotherapy will add only a limited life expectation for those patients. Thus, new biomarkers need to be evaluated in those conditions when patients present with early clinical symptoms such as the onset of diabetes, or even before the presence of solid or cystic lesions has been detected in ultrasounds or CT-scans.

MicroRNAs (miRNAs) are small non-coding RNAs that bind to complementary sequences to modulate the expression of targeted mRNAs. The seed sequence near CT scthe miRNA 5′ end base pairs with a target mRNA, thereby inducing deadenylation, which leads to altered translational regulation and decay (reviewed in [15]). MiRNAs can be detected at low levels, and miRNA expression profiles can be useful biomarkers. MiRNA expression profiling has revealed a general down-regulation of miRNAs in most tumors compared with normal tissues and even poorly differentiated tumors [16,17]. This suggests key roles for miRNA expression in the post-transcriptional silencing of targeted genes or the prevention of apoptosis by binding to promoter units involved in cell-cycle regulation. MiRNAs have been considered to be new biomarkers for general use in GI tumors. MiRNA expression profiles seem to classify human gastrointestinal cancer better than mRNA or protein-expression profiles [16]. Le Large et al. have pointed out that circulating miRNAs may be used as diagnostic biomarkers for ductal pancreatic adenocarcinoma, and circulating as well as local miRNAs have been proposed as potential diagnostic biomarkers [18,19]. Small miRNA panels that are currently being developed can be easily acquired and display a high specificity for the determination of malignant cells in solid tumors, cystic lesions, and chronically inflamed pancreatic tissue.

## 2. MiRNA Dysregulation in Blood Samples and Biliary or Pancreatic Duct Fluids Obtained from Patients with Pancreatic Ductal Adenocarcinoma (PDAC)

Blood samples from patients with pancreatic adenocarcinoma are easily available, and a very comprehensive study using NGS recently identified a miRNA panel (miR-20a, miR-21, miR-24, miR-25, miR-99a, miR-185, and miR-191) in blood samples that differentiated patients with pancreatic cancer from healthy controls remarkably well, with an AUC of 0.99 [20]. However, these data need to be interpreted with care since other tumors may be present too; microRNAs such as miR-21 are present in different tumor entities, and the origin of the type of cancer may differ to a great extent. Moreover, these data are in contrast to other results of recent studies [12,13,14]. These studies reported that the dysregulation of miRNAs miR-16, miR-27a-3p, miR-200a, and miR-159 in blood samples is associated with the increased presence of PDAC [21,22,23,24]. A quantitative and highly specific assay showed that miR-10b was dysregulated in biologic fluids and circulating exosomes [25]. MiRNAs in panels of circulating miRNAs in pancreatic juice have also been described as possible biomarkers of PDAC [26]. A miRNA meta-signature has been presented for PDAC [27]; among the various molecules described in that work, miR-21, miR-34a, and miR-155 were shown to be highly specific as diagnostic and prognostic discriminating biomarkers from human blood of patients with either chronic pancreatitis or PDAC [27]. MiR-198 and miR-217 have been shown to enable better differentiation of chronic pancreatitis and PDAC [28]. MiR-196a and miR-196b are potential serum biomarkers for the early detection of familial pancreatic cancer [29]. Details about all of these molecular markers in solid pancreatic tumors and other specimens are presented in Table 1 and an overview is given in Figure 1. Interestingly, a pilot study to develop a diagnostic test for PDAC based on the differential expression of selected miRNAs in plasma and bile has been developed but still lacks clinical approval [30,31]. MiRNA dysregulation has even been evaluated in salivary specimens, and one group has reported an association with PC [32]. To date, however, unambiguous and decisive predictive values that would allow the replacement of current tumor markers such as CA 19-9 have not been obtained.

With regard to the altered intracellular signal transduction procedures, details of the different effects of the dysregulated microRNAs are given in Table 1. For example, miR-21 has been reported to affect microsatellite instability (MSI) and was significantly associated with poor tumor differentiation [35]. MiR-196, in contrast, acts as a tumor suppressor by targeting the JAK2 oncogene [38,39,40], which in turn is downregulated in cancer cells and reduces cell viability via the caspase-mediated apoptosis pathway through the downregulation of PDK1. All of these molecules seem to act via different signal transduction pathways, and, thus, one common trunk of altered signal transduction cannot be identified in the pathogenesis of PDAC.

However, while NGS sequencing has led to a panel of microRNAs that seem to be frequently present in PDAC with AUC values of 0.99, most of these miRNAs cannot be claimed to be specific markers for PDAC since they have frequently been detected in other tumors. Consequently, the determination of this miRNA panel without clinical signs and risk constellations per se cannot prove the presence of PDAC, and, thus, the current use of these biomarkers as screening tools in blood or fluids appears to have only small clinical value. In contrast, in the context of the onset of diabetes mellitus frequently associated with PDAC presence, the evaluation of new biomarkers in blood or tissue could even gain special importance, enabling the detection tumors even earlier.

## 3. MicroRNA Dysregulation in Tumor Tissue of Patients with PDAC: Emerging Role for Risk Stratification in Patients with Newly Diagnosed Diabetes Mellitus and Hyperinsulinemia

The dysregulation of certain microRNA in pancreatic cancer as determined by the acquisition of tumor tissue is summarized in Table 1. For example, pancreatic tumors show miR-146a overexpression [36,37], and miR-146a dysregulation has been reported to promote tumorigenesis and metastasis. Moreover, miR-198 and miR-217 may be highly specific as diagnostic and differentiating biomarkers for chronic pancreatitis and pancreatic ductal adenocarcinoma [30]. Furthermore, miR-409 dysregulation has also been shown in PDAC; low expression was associated with poor outcomes, and miR-409 was reported to downregulate GAB-1 and antagonize PD-L1 action, a key molecule in immune checkpoint control [41,42]. Recent studies also revealed that different alterations in the signal transduction pathways are altered in response to the dysregulation of miR-409 and miR-490-3p. While miR-490 downregulates GAB-1 and antagonizes PD-L1 action, a key molecule in immune checkpoint control [41,42], miR-490-3p seems to suppress growth and metastasis in cell lines by targeting SMARCD1 [43].

The value of such molecular parameters may further be increased in the presence of clinical conditions that may be regarded as “sentinel conditions”. In this context, the onset of diabetes mellitus (DM) can be an early sign of pancreatic cancer. Thus, routine evaluations of elevated HbA1c levels in gastrointestinal patients could help to find those who are at risk of cancer development, especially when a secondary panel of valuable biomarkers is easily available. A meta-analysis of 88 studies [46] found a strong association between recently diagnosed DM and pancreatic cancer. These data suggest that patients with new-onset DM should not only be screened for pancreatic cancer by CT-scans and endoscopic ultrasound, but also with additional molecular tools that can be easily obtained. The onset of DM may, in part, be explained by the dysregulation of several growth factors, such as IGF-1 in patients with acromegaly, who are at risk of the development of pancreatic cancer as well as diabetes mellitus. IGF-1 is a key factor in the development of pancreatic cancer and is known to be subject to dysregulation by several miRNAs, such as miR-486 [47,48]. IGF-1 expression and hyperinsulinemia (acting alone or in parallel) associated with the onset of diabetes mellitus have been described as new and evolving factors that may explain the higher incidence of pancreatic cancer in Western populations [49]. Insulin and IGF-1 both promote growth and inhibit apoptosis in cancer cells through complex signaling. In addition to the insulin receptors (InsRs), the receptors of IGF (IGFRs) can homodimerize or form heterodimers with InsR, giving rise to a complex activation pattern through insulin as well as IGF-1 and IGF-2 [33,50].

Of great interest in this context, numerous studies have determined a possible link between insulin and immune evasion by pancreatic cancer cells. A recent molecular study showed that insulin induces the expression of the immune checkpoint regulator programmed death ligand 1 (PD-L1) [51]. Insulin-induced PD-L1 expression seems to allow pancreatic cancer cells to suppress the proliferation of CD8^+^ T-cells, leading to immune evasion [51]. In this context, another study found an association of very low miR-409 expression with very poor clinical outcomes in pancreatic cancer patients. The accelerated growth of the pancreatic cancer occurred via the downregulation of GAB1, a signaling factor involved in the control of apoptosis [52].

Interactions between long noncoding RNAs (lncRNAs) and miRNAs have recently been reported as new markers of risk stratification and a means to prevent pancreatic cancer [41]. By combining genome-wide association studies and functional data in almost 10,000 cases of PDAC, a significant association between the rs7046076 SNP and the risk of developing pancreatic ductal adenocarcinoma was detected, with a highly significant *p*-value (<0.0001). This SNP is located in the lnc-SMC2-1 gene and has been reported to disrupt the binding of the lncRNA with miR-1256, which regulates several genes involved in the cell cycle, such as CDKN2B. The CDKN2B region is pleiotropic, and its genetic variants have been associated with several human diseases, possibly though the interaction between lncRNA and miRNA [42]. A link to hyperinsulinemia could also be possible, linking HbA1c expression panels with the dysregulation of miRNAs such as miR-1246, miR-1256, and miR-1290 [44,45,53]. Thus, these miRNA panels could be introduced into clinical practice to help predict whether pancreatic solid masses should be regarded as either malignant and needing further surgical resection or as benign tumors where (only) subsequent controls should be performed.

## 4. MicroRNA Expression in Patients with Cystic Pancreatic Lesions, as Well as Pre- and Malignant Cystic Pancreatic Cancer Type

The incidence of cystic pancreatic tumors that originate from the epithelium has been summarized in an initial but very clear study from France, conducted on over 500 patients. In that work, four different entities were described: benign serous-cystic adenoma (SCA, 32%), premalignant mucinous cystadenoma (MCA, 29%), malignant tumors described as cystadenocarcinoma (MCAC, 5%), and intraductal papillary mucinous tumors involving the main pancreatic duct (IPMN, 11%) [54]. So far, the study has found that IPMNs are the most common premalignant lesions; they are classified based on the involvement of the pancreatic ductal system as either main-duct (MD) IPMN, branch-duct (BD) IPMN, or the combined type. The malignant development of the lesions is frequent, especially in elderly patients, while benign SCAs occur in middle-aged women and are typically microcystic lesions with a size between 1 and 12 cm, with a diameter around 3 cm. Mucinous-cystic adenomas (MCA) MCA are more common in middle-aged women (female:male 9:1, 40 to 50 years of age), and 85% are in the pancreatic body and tail area with a size between 2 and 15 cm. In a small percentage, progression to adenocarcinoma is observed. Age is a major risk factor when evaluating the dignity of such cystic pancreatic lesions, as is the expression of the tumor marker carcinoembryonic antigen (CEA) in cyst fluids. A recent meta-analysis examined the predictive value of tumor markers in cyst fluids in discriminating between malignant and pre-malignant lesions [9,10]. In a pooled analysis of 12 studies with 450 patients [9], a low amylase concentration of 250 ng/mL was typical for an SCA or MCA (the sensitivity was 44%, the specificity 98%), and a CEA value above 800 ng/mL was typical for a malignant process [9]. Cytology was positive in about half of the cases and detected malignant cells in the mucinous-cystic tumors in 48% of the cases. The authors came to a recommendation that resection should be attempted in most patients with cystic pancreatic tumors and high expression levels of CEA tumor markers, but this decision is still controversial. In parallel, the liquid biopsy analysis of these cyst fluids to examine CEA expression has been shown to have a sensitivity of only 59% to 67% and a specificity of 83% to 91% in detecting mucinous cysts [55].

The 5-year risk of developing dysplasia was 63% in patients with the main duct type compared with 15% in patients with the side-branch type. It is particularly worth mentioning that patients with the “branch-duct type” have a significantly reduced risk of developing carcinomas, which could justify a non-surgical approach in old age. Incidental or bland pancreatic cysts present the greatest problems in terms of differential diagnosis and therapeutic approach. Tada et al. examined the risk of developing carcinoma in such pancreatic cysts in a recent study, in which 197 patients were observed for 5 years. Carcinomas, including five intraductal carcinomas and two IPM tumors, were found in seven patients. There was thus a carcinoma incidence of 0.95% in this group, which was significantly higher than in the normal collective [14,56,57,58,59].

Recent pilot studies have, therefore, investigated miRNA expression profiles in pancreatic cysts and have found differences in expression profiles that allow the discrimination between premalignant and malignant cysts [60]. For example, miR-21 levels were found to predict cancer development in mucinous precursor lesions, and miR-21 dysregulation can be found in numerous GI tumors [60,61]. Other works have described miRNAs as reliable biomarkers in cyst fluid, but the number investigated was quite small [60,62].

In a recent work investigating 70 patients [63], several miRNA panels were described that differentiated among the lesions. One panel divided benign SCA lesions from malignant cysts with very good sensitivity and specificity; a second panel distinguished MCN from SCA, BD-IPMN, MD-IPMN, and PDAC with sensitivity and specificity of 100%; and another panel differentiated PDAC from IPMN. Importantly, the authors described a miRNA classifier distinguishing between SCA and mucinous pancreatic cystic neoplasms with 90% sensitivity and 100% specificity. This panel was composed of miR-31-5p, miR-483-5p, miR-99a-5p, and miR-375 [63]. In this very comprehensive study, the authors also found the dysregulation of 10 different miRNA, including miR-135a/b, miR-200a/b/c, miR-224, miR-363, miR-429, miR-708, and miR-885-5p, in main-type IPMT fluid cysts [63]; these miRNAs were not detected in benign lesions such as SCA and MCN. In a further study, six miRNAs were detected at significantly higher levels in cyst fluid from IPMC than in cyst fluid from IPMA. They included miR-711, miR-3679-5p, miR-6126, miR-6780b-5p, miR-6798-5p, and miR-6879-5p, differing notably from the above group, perhaps due to the small sample size [64]. Wang et al. identified miRNAs that have decreased malignant cyst fluid contents compared with benign cysts (miR-451a and miR-4284) [65]. Thus, these expression profiles seem to help clinicians decide whether a cystic formation should be resected or not, supporting the high clinical importance of miRNA panels. Table 2 summarizes the current knowledge of deregulated miRNA in cystic pancreatic lesions, including the molecular mechanisms involved and tumor suppressor or oncogenic factors, as well as the statistical significance factors reported in the studies.

### Perspective: MicroRNA Dysregulation as Differentiating Parameters for Pancreatic Tumors and Cystic Lesions: Diverging Expression and Signal Transduction

So far, the determination of the biomarker CA 19-9 in human serum is widely used as an indicator and course predictor of pancreatic cancer, but this value has raised several concerns including not only the abnormal expression in various benign conditions (chronic pancreatitis, liver cirrhosis or cholangitis), but also the absence in up to 12% of the Caucasian population, leading to a poor predictive value in detecting PDAC [67]. Moreover, this biomarker has a very poor value in differentiating solid from cystic pancreatic lesions.

Therefore, complex miRNA expression profiles and certain “panels” have been described in various gastrointestinal cancers including PDAC at different stages, and dysregulated miRNA expression patterns exists in PDAC in tumor tissues, plasma, sputum, and stool samples of patients; therefore, these markers seem to have a high potential to serve as minimally invasive screening and novel tools for diagnostic evaluations [35]. Using blood samples, a well-designed study using NGS identified a miRNA panel (miR-20a, miR-21, miR-24, miR-25, miR-99a, miR-185, and miR-191) that differentiated patients with pancreatic cancer from healthy controls remarkably well, with an AUC of 0.99 [20]. Circulating miRNAs in serum, however, have been investigated only in small numbers in this field, and, thus, the observed alterations in circulating miRNAs may represent only sporadic observation with little consensus among multiple other studies carried out by different groups and dysregulation in other different tumors, thereby disabling the use as biomarkers for clinical utility at present. This field may be altered soon when better clinical data in the course of the disease becomes available.

In solid pancreatic tumor tissue, the dysregulation of miR-146, miR-196a/b, miR-198, miR-217, miR-409, and miR-490 has been detected in PDAC and may predict cancer presence in PDAC patients. These tissue biomarkers may reach a special value in the context of the clinical conditions of newly diagnosed diabetes mellitus and hyperinsulinemia. A panel of three microRNAs, i.e., MiR-196, miR-486, and miR-1290 may thus be of special clinical interest in the context of patients presenting with an unclear pancreatic tumor but also reporting metabolic problems.

In cystic pancreatic tumors, as well as premalignant cystic pancreatic lesions, such as intraductal papillary neoplasia (IPMN) and mucinous-cystic adenomatous cysts (MCAC), the dysregulation of a panel of four microRNAs, i.e., miR-31-5p, miR-483-5p, miR-99a-5p, and miR-375, has been found to be of high clinical value in differentiating benign from malignant lesions (summary is given in Table 2).

Of special interest in this context may be that the signal transduction pathways induced in patients with cystic pancreatic tumors seem to be entirely different from patients with PDAC. For example, the fact that miR-31-5p has been reported to be dysregulated in cystic pancreatic lesions has also been investigated in colorectal cancer patients, and it was demonstrated that the structural regulation of the miR-31-5p-dystrophin axis was altered in CRC patients. In CRC, altered dystrophin (DMD, a musculo-cellar protein) expression was linked to the onset and progression of cancer, including myogenic tumors and even non-myogenic tumors, and it is considered as a new regulatory factor in tumor development and a new prognostic factor for tumor progression and the survival of CRC patients. Thus, the dysregulation of the miR-31-5p-DMD axis may be a novel biomarker for predicting the development and prognosis not only of sporadic early-onset colorectal cancer, but also of cystic pancreatic neoplasms [37]. In line with these observations, the miR-483-5p that was detected in cystic pancreatic lesions seems to act via the down-regulation of the TGF-ß pathway and induces altered signals for fibrosis and malignant transition [38], while miR-99a-5p seems to alter the pathways in epithelial-mesenchymal transition, as reported in cervical squamous cell carcinoma by targeting CDC25A/IL6 pathways [39]. Thus, these signaling pathways seem to be entirely different from PDAC and, thus, may also be regarded as differentiating markers.

Patients with newly diagnosed diabetes mellitus in the context of PDAC may be considered to be a special risk group in which newly described biomarkers might help to facilitate a tumor diagnosis even before a malignant histology is available. Several studies have been published that describe the special role of miRNA dysregulation in the onset and progress of diabetes [68]. It appears that this dysregulation reflects a process that has diverging intracellular signal transduction pathways. Specifically, it has been reported that microRNA dysregulation seems to be of special importance for pancreatic ß-cell apoptosis and even malignant transformation. Through the PI3K pathway, miR-375 seems to play a special inhibitory role in β-cell proliferation and survival [69]. Furthermore, MiR-21 seems to be involved in the cellular machinery regulating β-cell turnover [70,71]. miR-21 was found to induce an increase in β-cell proliferation [70], presumably by acting on cell-cycle genes involved in checkpoint regulation of the G1 phase. Moreover, the proinflammatory cytokine NFκB pathway has been reported to up-regulate the expression of miR-21, which may play a role in increasing NO synthesis and βl apo-cell apoptosis [70]. Moreover, another study, which used overexpression of miR-21 in cell lines, did not find any effect on cell survival but reported that decreasing miR-21 expression promoted apoptosis [71]. Finally, miR-200 was shown to be strongly linked to β-cell pathology in both type 1 and type 2 diabetes mellitus. Specifically, miR-200 suppressed the anti-apoptotic and stress-resistance pathways of β-cell heat shock protein, and Xiap, a caspase inhibitor, and miR-200 also facilitated the activation of the tumor suppressor protein Trp53, fostering the expression of pro-apoptotic genes [72]. The dysregulation of these interesting microRNAs might not only be associated with the onset of diabetes, but also with a malignant transformation at earlier stages.

Taken together, a panel of the above-described microRNAs (miR-486, miR-196, and miR-1256, as well as miR-375, miR-21, and miR-200) might help to described altered ß-cell functions in the early onset of pancreatic cancer development, and, thereby, the new aspects might help to detect this kind of cancer earlier. At the time point when these conditions are diagnosed, a simple and easily available biomarker would help to facilitate immediate treatment, and modern concepts favoring the role of neoadjuvant chemotherapy even 6 months before surgery and a solid tumor mass has developed may become the standard treatment within the next few years. Currently, at the point when solid tumors are detected, it may already be too late to achieve curative treatment in PDAC, and, thus, understanding the differentially dysregulated microRNAs might help to detect tumors even at an earlier stage.

## Figures and Tables

**Figure 1 cells-11-02374-f001:**
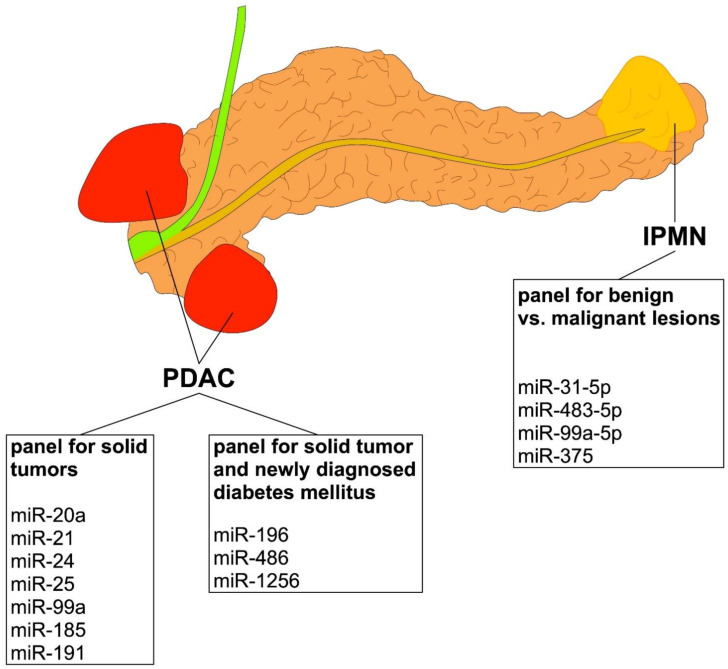
Overview of different panels to compare between PDAC and IPMN pancreas lesions.

**Table 1 cells-11-02374-t001:** Overview of dysregulated miRNAs involved in pancreatic ductal adenocarcinoma, as determined in different sources.

Type of MiRNA	Role in Pancreatic Ductal Adenocarcinoma and Potential Molecular Targets	Origin of Material	References
miR-20a, miR-21, miR-24, miR-25, miR-99a, miR-185, and miR-191	Panel of different microRNAs determined in serum of PDAC patients indicating cancer presence with AUC above 0.99.	Blood sample/serum	[20], also confirmed in [33,34].
miR-196a/b	miR-196a and -196b can act as potential biomarkers for the early detection of familial pancreatic cancer in serum.	Upregulated in serum	[31]
miR-217	miR-217 has been shown to be a prognostic biomarker for chronic pancreatitis and pancreatic ductal adenocarcinoma.	Downregulated in serum, also tumor tissue	[30]
miR-21	Affects microsatellite instability (MSI); significantly associated with poor tumor differentiation [35]. miR-21 overexpression promotes cancer-cell growth, invasion, and migration in vitro; prognostic marker for local invasion and lymph node metastasis. Not specific for a defined GI tumor, also downregulated in gastric or colon cancer [33,34].	Overexpressed in tumor tissue	[33,34,35].
miR-146a	Pancreatic tumors show miR-146a overexpression. miR-146a dysregulation may promote tumorigenesis and metastasis.	Overexpressed in tumor tissue	[36,37]
miR-198	miR-198 may be highly specific as diagnostic and differentiating biomarker for chronic pancreatitis and pancreatic ductal adenocarcinoma.	Downregulated in inflamed pancreatic tissue/tumor tissue	[30]
miR-375	Acts as a tumor suppressor by targeting the JAK2 oncogene. Downregulated in cancer cells, reduces cell viability via the caspase-mediated apoptosis pathway through downregulation of PDK1.	Downregulated in tumor tissue	[38,39,40]
miR-409	Downregulates GAB-1 and antagonizes PD-L1 action, a key molecule in immune checkpoint control. Low expression associated with poor outcomes.	Downregulated in tumor tissue	[41,42]
miR-490-3p	miR-490-3p suppresses growth and metastasis in cell lines by targeting SMARCD1.	Downregulated in tumor tissue, tumor suppressor under normal conditions	[43]
**MicroRNA dysregulation during special clinical conditions**	**Role during onset of diabetes and hyperinsulinemia (“Sentinel markers”)**	**Origin of** **Material**	**References**
miR-196	miR-196b expression levels in pancreatic cancer cells were significantly higher than those of cancer stroma and correlates with the long-term survival. It can activate the AKT signaling pathway, which is involved in the development and treatment of type 2 diabetes	Upregulated in tumor tissue, oncogenic potential	[34]
miR-486	miR-486 is upregulated in tumor tissue of PDAC patients with newly diagnosed diabetes mellitus and promotes the proliferation of pancreatic cancer cells.	Upregulated in tumor tissue, oncogenic potential	[36]
miR-1256	miR-1256 regulates several genes involved in the cell cycle, such as CDKN2B. The CDKN2B region is pleiotropic, and its genetic variants have been associated with several human diseases, possibly through the interaction between lncRNA and miRNA.	Upregulated in tumor tissue, oncogenic potential	[44,45]

**Table 2 cells-11-02374-t002:** MiRNA expression in pancreatic cysts and malignant pancreatic tissue. Comparison of dysregulation is shown in cysts or premalignant vs. malignant tissue. Relevant *p*-values for discrimination are presented between different entities and outlined depending on the significance levels (see References [63,64,65]).

Type of MiRNA	Type of Dysregulation in Different Kinds of Cystic Pancreatic Lesions	*p*-Value
miR-21	Mucinous vs. Non-mucinous Cysts: significantly associated with poor tumor differentiation. MiR-21 affects microsatellite instability (MSI) and miR-21 overexpression promotes cancer cell growth, invasion, and migration in vitro [35]. Prognostic marker for local invasion and lymph-node metastasis [33,34].	<0.0025
miR-31-5p	SCA vs. MCN/PDAC/IPMN Oncogenic effects in cystic pancreatic lesions and colorectal cancer patients that include structural dysregulation of the miR-31-5p-dystrophin axis. Altered dystrophin (DMD, a musculo-cellular protein) expression promotes myogenic tumors and even non-myogenic tumors, considered as a new regulatory factor in tumor development and a new prognostic factor for tumor progression. Dysregulation of miR-31-5p-DMD axis may be a novel biomarker for predicting the development and prognosis not only of sporadic early-onset colorectal cancer, but also of cystic pancreatic neoplasms [37].	<0.0125
miR-483-5p	SCA vs. MCN/PDAC/IPMN Downregulated in tumors, tumor-suppressor quality. miR-483-5p detected in cystic pancreatic lesions seems to act via down regulation of the TGF-ß pathway and induces altered signals for fibrosis and malignant transition [38].	<0.0125
miR-99a-5p	SCA vs. MCN/PDAC/IPMN Oncogenic effects, alters pathways in epithelial mesenchymal transition, as reported in cervical squamous cell carcinoma by targeting CDC25A/IL6 pathways [39].	<0.0125
miR-375	SCA vs. MCN/PDAC/IPMN Downregulated in tumor cells. Acts as a tumor suppressor by targeting the JAK2 oncogene. Downregulated in cancer cells, reduces cell viability via the caspase-mediated apoptosis pathway through downregulation of PDK1.	<0.0125
	MicroRNA with special importance for BD-IPMN	
miR-200a/b/c	BD-IPMN vs. MCN Downregulated in tumors and especially in lung metastasis of colon cancer, strong suppressive effect on cell transformation [40].	<0.01
miR-224	BD-IPMN vs. MCN Oncogenic potential, promotes tumor progression in small cell lung cancer [43].	<0.01
miR-363	BD-IPMN vs. MCN Tumor suppressor gene. Overexpression of miR-335 and miR-363 decreased tumorigenicity as measured by clonogenicity, anchorage-independent growth, and metastasis determined by cell invasion assay and liver metastasis in vivo [66].	<0.01
	**Dysregulated microRNAs with unknown** **significance and function in cystic pancreatic lesions**	
miR-708	BD-IPMN vs. MCN	0.01
miR-429	BD-IPMN vs. MCN	0.01
miR-3679-5p	IPMC vs. IPMA	0.048
miR-6126	IPMC vs. IPMA	0.016
miR-711	IPMC vs. IPMA	0.016
miR-6798-5p	IPMC vs. IPMA	0.048
miR-6879-5p	IPMC vs. IPMA	0.048

BD-IPMN, branch duct-intraductal papillary mucinous neoplasm; MCN, mucinous cystic neoplasm; SCA, serous cystadenoma; IPMC, intraductal papillary mucinous cyst; IPMA, intraductal papillary mucinous adenoma; PDAC, pancreatic ductal adenocarcinoma.

## Data Availability

Not applicable in this review.

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
