# Peer review of "MicroRNAs as Indicators of Malignancy in Pancreatic Ductal Adenocarcinoma (PDAC) and Cystic Pancreatic Lesions"

_cells, 2022, doi:10.3390/cells11152374_

Round 1
Reviewer 1 Report
In this review by Prinz et al the authors have reviewed previous reports related to MicroRNAs as Indicators of Malignancy in Pancreatic Ductal Adenocarcinoma (PDAC) and Cystic Pancreatic Lesions. The main problem about such reviews, specifically focussing on pancreatic cancer are- the previous literature is flooded with such reviews on pancreatic cancer. Thus another such reviewdoes not provide any new insight as the role of miRNAs in pancreatic cancer. It may be important that the authors explain the reasons for similarites and differences in miRNA profiles between tissues and body fluids . Also they may explain, with concrete examples, the observed differences in miRNA profiles between different studies. Unless they provide discussion based on isuch mechanistic insight the reviews become just another repository of reports from previous literature. Also the review is poorly written and may need lot of improvement in the style which it has been written
Author Response
In this review by Prinz et al the authors have reviewed previous reports related to MicroRNAs as Indicators of Malignancy in Pancreatic Ductal Adenocarcinoma (PDAC) and Cystic Pancreatic Lesions. The main problem about such reviews, specifically focussing on pancreatic cancer are- the previous literature is flooded with such reviews on pancreatic cancer. Thus another such reviewdoes not provide any new insight as the role of miRNAs in pancreatic cancer. It may be important that the authors explain the reasons for similarites and differences in miRNA profiles between tissues and body fluids . Also they may explain, with concrete examples, the observed differences in miRNA profiles between different studies. Unless they provide discussion based on isuch mechanistic insight the reviews become just another repository of reports from previous literature. Also the review is poorly written and may need lot of improvement in the style which it has been written.
Answer to reviewer 1: This review has a special emphasis on the importance of microRNA deregulation especially in patients with newly diagnosed diabetes mellitus, as well as cystic pancreatic lesions. The manuscript describes important differences of these patients and tumor entities when compared to tumor tissue or blood samples obtained from patients with pancreatic adenocarcinoma. To our knowledge, this is a first work compared all three entities.
IN addition, we have modified the manuscript according tot he reviewers comments, and added more details regarding the function of microRNA in TABLE 1 and TABLE 2. Also, a new paragraph is added at the end describing the role of microRNA as biomarkers for PDAC.
Reviewer 2 Report
Well-written manuscript
Author Response
Thank you for the nice review
Reviewer 3 Report
MiRNAs are non-coding small RNAs that regulate the expression of multiple proteins in the post-translation process and have promise as biomarkers, prognostic agents, and as advanced pancreatic therapies. In this review, the author summarized dysregulated miRNAs in pancreatic ductal adenocarcinoma and cystic pancreatic lesions. The functional overview of these microRNAs would be helpful for us to understand their functions in PDAC. While the authors have made an amount of effort for data analysis, there are some points that need to be adequately addressed.
1) What are potential targets of these dysregulated miRNAs? The potential target mRNAs/lncRNAs of the examples of dysregulated miRNAs should also be shown in Table1.
2) miRNAs play a major role in carcinogenesis, falling into two categories: tumour suppressor miRNAs, and oncogenic miRNAs. The authors also should include above category information in Fig1 or in Table1.
3) Could the authors also summarize the prognostic role of these miRNAs? For example, which miRNA can serve as poor survival biomarkers? Which miRNA was better survival markers?
4) The therapeutic potential of miRNAs in PDAC was more interesting. Could the author discuss more about the potential strategies to develop miRNA-based therapeutics in PDAC.
5) Line 76, 300, and 307 contain multiple blank spaces, kindly rectified.
6) Line 38 and 67 were not a complete sentence, kindly rectified.
In summary, miRNAs could serve as potential biomarkers, prognostic markers and clinical targets for PDAC. This review is valuable in the current miRNA research field in PDAC.
Author Response
please see attached comments.

Round 2
Reviewer 1 Report
Although the authors have improved this manuscript to some extent it may still not reach to the level that may be considered significant contribution over previously published reviews. As suggested in the previous review, It may be important that the authors explain the reasons for similarites and differences in miRNA profiles between tissues and body fluids . Also they may explain, with concrete examples, the observed differences in miRNA profiles between different studies. Although they have reported miRNAs that may be of significance in pancreatic cancer under certain clinical conditions(e.g. diabetes), infomations like miRNAs that might be very specific only in pancreatic cancers (developed under that conditions) but but in pancreatic cancer without such conditions, may be more important. Thus the authors may need to put in significantmore efforts to put together such infomation to make the manuscript publishable.
Author Response
Reply to the questions raised by Reviewer 1 in the R2 review process.
Thank you very much fort he critical reappraisal and feedback to the overall impact of our manuscript. We understand that our current work appears only as a small part within the large amount of data given so far focussing on microRNA expression in pancreatic cancer. However, from a clinical point of view, the approach to patients with pancreatic adenocarcinoma is in most cases delayed so that the majority of patients is diagnosed when the tumor has frequently metastasized and despite modern treatment, surgery and chemotherapy will add only a limited life expactation in those patients. Thus, new biomarkers need to be evaluated in those conditions when patients present with early clinical symptoms such as the onset of diabetes, or even before the presence of solid or cystic lesions has been detected in ultrasound or CT scans (this passage has been added to setion 1 – introduction).
Thus, we have answered the following issues raised in your review as follows:
- Q: Describe important differences between body fluids and tissue.
A: However, while NGS sequencing has led to a panel of microRNAs that seem to be frequently present in PDAC with AUC values of 0.99, most of these miRNAs can not be claimed specific markers for PDAC since they have frequently been detected also in other tumors. Consequently, the determination of this miRNA panel without clinical signs and risk constellations per se cannot proof the presence of PDAC and thus, current use of these biomarkers as screening tools in blood or fluids appears only small clinical value. In contrast, in the context of the onset of diabetes mellitus frequently associated with PDAC presence, the evaluation of new biomarkers in blood or tissue could even gain special importance, enabling to detect tumors even earlier (Added at the end of section 2.)
- Q: Differential expression of microRNAs- Describe differences and concordance between various studies.
A: This critique has already been integrated into the manuscript and in details in Table 1.
- Q: Describe the differential role of miR-expression during clinical conditions like the onset of diabetes and the absence of diabetes.
Answer:
Patients with newly diagnosed diabetes mellitus in the context of PDAC may be considered as a special risk group in which newly described biomarkers might help to facilitate a tumor diagnose even before a malignant histology is available. Several studies have been published that describe a special role of miRNA dysregulation in the onset and progress of diabetes [68]. It appears that this dysregulation reflects an own process that has diverging intracellular signal transduction pathways. Specifically, it has been reported that microRNA dysregulation seems to be of special importance for pancreatic ß-cell apoptosis and even maligant transformation. Through the PI3K pathway, miR-375 seems to play a special inhibitory role in β-cell proliferation and survival [69]. Furthermore, MiR-21 seems to be involved in the cellular machinery regulating β-cell turnover [70,71]. miR-21 was found to induce an increase in ß-cell proliferation [70], presumably by acting on cell cycle genes involved in checkpoint regulation of the G1 phase. Also, the proinflammatory cytokine NFκB pathway has been reported to up-regulate the expression of miR-21, which may play a role in increasing NO synthesis and βl apo-cell apoptosis [70]. Also, another study which used overexpression of miR-21 in cell lines did not find any effect on cell survival, but reported that decreasing miR-21 expression promoted apoptosis [71]. Finally, miR-200 was shown to be strongly linked to β-cell pathology in both type 1 and type 2 diabetes mellitus. Specifically, miR-200 suppressed the anti-apoptotic and stress-resistance pathways of β-cell heat shock protein, and Xiap, a caspase inhibitor, and miR-200 also facilitated the activation of the tumor suppressor protein Trp53, fostering the expression of pro-apoptotic genes [72]. Dysregulation of these interesting microRNAs might be associated with the onset of diabetes, but also a malignant transformation at earlier stages.
Taken together, a panel of the above described microRNAs (miR-486, miR-196, miR-1256 as well as miR-375, miR-21 and miR-200) might help to described altered ß-cells function in the early onset of pancreactic cancer development and thereby, the new aspects might help to detect this kind of cancer earlier. At the time point when these conditions are diagnosed, a simple and easily available biomarker would help to facilitate immediate treatment, and modern concepts favouring the role of neoadjuvant chemotherapy even 6 months before surgery and a solid tumor mass has developed may become the standard treatment within the next years. Currently, at the point when solid tumors are detected, it may already be too late to achieve curative treatment in PDAC, and thus, understanding of the differentially dysregulated microRNAs might help to detect tumors even at an earlier stage.
(added in the final section at the end of the ms).

This manuscript is a resubmission of an earlier submission. The following is a list of the peer review reports and author responses from that submission.
Round 1
Reviewer 1 Report
The manuscript entitled “MicroRNAs as Indicators of Malignancy in Pancreatic Ductal Adenocarcinoma (PDAC) and Cystic Pancreatic Lesions” by Prinz, et al reviews current knowledge of dysregulation of microRNA expression in PDAC and cystic pancreatic lesions. Overall, the manuscript is a sort review which summarizes the emerging microRNA markers for diagnosis and prognosis of PDAC and pancreatic cystic lesion. It covers some topics in the selected area. However, there are several concerns, which need to be further addressed as follows.
1. Authors should add figures and contents to help readers to understand the importance of deregulated microRNAs in PDAC and cystic lesion development and formation.
2. As authors mentioned, liquid biopsy is a new approach for diagnosis of pancreatic lesions. In section 2, authors should provide more contents to discuss different deregulated microRNA in blood samples, salivary specimens, stool samples, biliary or pancreatic duct fluids in different subsections. Authors need to discuss the importance of these deregulated microRNAs in different specimens.
3. Authors could add more information on how these deregulated microRNAs alter cellular signal transduction, causing different pancreatic lesions.
4. Authors should add a “conclusion” section to summarize what microRNAs they believe are critical for the diagnosis and prognosis of pancreatic malignant lesions in liquid biopsy for clinic use.
Reviewer 2 Report
The present mini-review does not add anything in the already published literature
Reviewer 3 Report
In this article by Prinz et al the authors have reviewed the work on MicroRNAs as Indicators of Malignancy in Pancreatic Ductal Adenocarcinoma (PDAC) and Cystic Pancreatic Lesions. There have been considrable number of reviews published that that have covered this subject matter as focused specially on pancreatic cancer or on pancreatic cancer along with other GI cancers in the same review. Thus, this review does not provide significantly more information than these other reviews provide. Also the authors have referred miRNAs or miRNA panels that were identified using different techniques which substantially different from each other. So unless they can provide explanations for such differences it may not be more meaningful information. Thus, this reviewer does not think the work may be of high priority deserving publication